# Effectiveness of Focal Muscle Vibration in the Recovery of Neuromotor Hypofunction: A Systematic Review

**DOI:** 10.3390/jfmk8030103

**Published:** 2023-07-25

**Authors:** Luigi Fattorini, Angelo Rodio, Guido Maria Filippi, Vito Enrico Pettorossi

**Affiliations:** 1Department of Physiology and Pharmacology “V. Erspamer”, School of Medicine, Faculty of Medicine and Surgery, Sapienza Università di Roma, L.go A. Moro 5, 00185 Rome, Italy; luigi.fattorini@uniroma1.it; 2Department of Human Sciences, Society and Health, University of Cassino and Southern Lazio, Loc. Folcara, 03043 Cassino, Italy; 3Department of Neuroscience, School of Medicine, Faculty of Medicine and Surgery, Università Cattolica del Sacro Cuore, Largo F. Vito 1, 00168 Rome, Italy; 4Fondazione Policlinico Universitario A. Gemelli IRCCS, Largo A. Gemelli 8, 00168 Rome, Italy; 5Department of Medicine and Surgery, Human Physiology Section, Università degli Studi di Perugia, Piazzale Gambuli 1, 06129 Perugia, Italy; vito.pettorossi@unipg.it

**Keywords:** muscle vibration, proprioception, muscle spindle, proprioceptive training, rehabilitation, motor function

## Abstract

Adequate physical recovery after trauma, injury, disease, a long period of hypomobility, or simply ageing is a difficult goal because rehabilitation protocols are long-lasting and often cannot ensure complete motor recovery. Therefore, the optimisation of rehabilitation procedures is an important target to be achieved. The possibility of restoring motor functions by acting on proprioceptive signals by unspecific repetitive muscle vibration, focally applied on single muscles (RFV), instead of only training muscle function, is a new perspective, as suggested by the effects on the motor performance evidenced by healthy persons. The focal muscle vibration consists of micro-stretching-shortening sequences applied to individual muscles. By repeating such stimulation, an immediate and persistent increase in motility can be attained. This review aims to show whether this proprioceptive stimulation is useful for optimising the rehabilitative process in the presence of poor motor function. Papers reporting RFV effects have evidenced that the motor deficits can be counteracted by focal vibration leading to an early and quick complete recovery. The RFV efficacy has been observed in various clinical conditions. The motor improvements were immediate and obtained without loading the joints. The review suggests that these protocols can be considered a powerful new advantage to enhance traditional rehabilitation and achieve a more complete motor recovery.

## 1. Introduction

Neuromotor hypofunction refers to a condition in which an individual has difficulty or limitations controlling and coordinating a movement and can affect various aspects of motor skills, from fine motor skills (such as writing, eating, buttoning a shirt) to gross motor skills (such as lifting weights, walking or running). The functional impairment consists of ‘negative symptoms’, such as asthenia, weakness, poor motor coordination and fatigue. These functional deficits can be induced by various neuromuscular diseases and by musculoskeletal trauma during daily, sporting and professional activities. For the latter, several individual characteristics are possible risk factors, such as age, poor fitness level, comorbidities, etc. Often, motor function may be restored after a long period of rehabilitation by providing impairment-specific intervention protocols. It should be noted that the motor and functional level obtained at the end of a rehabilitation period may not be sufficient to ensure full motor recovery. This is particularly true in sports, where full recovery of coordination and conditioning skills can be achieved with an additional conditioning period. An obvious limitation of exercise-based rehabilitation protocols is imposed by the individual’s functional residual, as well as expected individual compliance.

Optimisation of rehabilitation [1,2] is an area of study that always offers new topics, as research is constantly looking for protocols that reduce duration, avoid relapse and ensure full recovery of motor functions. A new direction of research suggests the possibility of acting on motor control in addition to or as an alternative to traditional exercise [3].

Recent reviews report evidence that sustained activation of the proprioceptive system can induce immediate improvements in motor abilities in healthy subjects [4,5,6]. Indeed, literature data show that improvements in muscle strength, motor task readiness, muscle power and movement fluidity and coordination can be achieved with proprioceptive stimulation, suggesting that this intervention be included in traditional rehabilitation programmes to improve subsequent functional motor deficits. These possibilities find important support in the consideration that motor movements and performance are largely based on proprioceptive input, upon which motor planning and execution are built and controlled [3]. On the other hand, proprioception deficits lead to profound alterations in motor execution, altering the control of postural reflexes, muscle tone and motor behaviour [7], as well as the spatial and temporal aspects of movement [8,9]. Although motor execution is supported by different sensory modalities (auditory, visual and tactile systems are involved), the proprioceptive system is considered the main source of information to plan and perform elementary and complex motor tasks. Improved proprioceptive processing could improve the accuracy of motor execution, as well as motor efficiency (i.e., muscle strength, fatigue), the latter being profoundly influenced by coordination. Therefore, an improvement of proprioceptive input could be a reasonable approach to improve or restore motor function. Furthermore, training the proprioceptive system does not require loading the skeletal system, avoiding one of the most relevant obstacles and allowing rehabilitation procedures to be anticipated, possibly preventing possible relapses. 

The literature shows that sensory modalities can be improved by applying repeated sequences of a specific vibratory stimulation (repeated focal muscle vibration, RFV), which has been reported to selectively activate the neuromuscular spindle [10]. This prolonged artificial activation of the proprioceptive system can induce persistent after-effects in proprioceptive circuits and increase the sensitivity and discrimination of the sensory system, as occurs in the tactile, visual and auditory systems after repetitive activation [11]. Best and Dinse attributed the after-effects to a form of long-term potentiation, like plastic changes in neural circuits, related to the specific stimulated modality. Specifically, the most effective focal muscle vibration protocol for neuromuscular spindle activation consists of mechanical vibration with sinusoidal sequences of micro-stretching-shortening of individual muscles, with a range of 0.2–1 mm, at approximately 100 Hz [12,13]. A recent review [6] shows that several studies have documented how adequate and prolonged proprioceptive stimulation can persistently increase the motor performance of healthy subjects. Although different protocols were applied in the analysed studies, positive results were obtained by applying the aforementioned stimulation parameters several times on the same day and repeated on consecutive days. On this basis, we suggest that RFV could also be useful in recovering motor impairment. The purpose of this review is to examine whether an intervention adopting RFV can be successfully applied in cases of poor motor function and whether, therefore, this proprioceptive stimulation might be inserted in rehabilitation protocols to obtain better findings. Moreover, these synergic results are in the direction of a shorter in time and more powerful reconditioning of coordinative and conditional abilities.

The review is limited to studies of individuals with neuromuscular hypofunction without spasticity, dyssynergy and dystonia. This limitation is justified by the desire to test whether the effect of focal vibration in individuals in whom motor activity should be enhanced is in line with the effect observed in healthy subjects. Therefore, subjects with hypertonia were excluded due to different levels of their circuit excitability.

## 2. Materials and Methods

The Preferred Reporting Items for Systematic reviews and Meta-Analyses (PRISMA) guidelines were followed in this review (Figure 1) [14].

### 2.1. Data Sources

A systematic literature search was conducted from January 1985 to March 2023 in the online databases PubMed, Web of Science and The Cochrane Library. The following Medical Subject Headings (MeSH) of the United States National Library of Medicine (NLM) and search terms were included in our Boolean search syntax: ((focal OR local OR segmental) AND vibration). The search was limited to the English language, human species, full-text availability and only original studies. Other relevant papers were identified through manual searching of potential papers based on the author’s knowledge.

### 2.2. Selection Criteria

Two reviewers (LF, GMF) extracted relevant data according to a structured script from each study. The structured script included study design, sample characteristics (sample size, genre), experimental and control group characteristics, outcome measure and timing of results. Inclusion criteria were decided by the consensus statements of these two reviewers. In cases where LF and GMF did not reach an agreement on the inclusion of an article, AR and VEP were contacted. Inclusion criteria were selected by (a) population: subjects showing stable negative deficits of muscle function; (b) intervention: treatment adopting localised mechanical vibration; (c) outcome: neuromuscular parameters regarding conditional abilities. Exclusion criteria were the absence of spasticity, rigidity, dystonia and dyssynergia.

### 2.3. Study Eligibility

Studies were excluded if they (a) analysed outcomes in a short follow-up period (<24 h) immediately after a single stimulation day; (b) the administration of the stimulus involved, at the same time, many different muscles; (c) did not present an original investigation (reviews or proceedings); (d) did not publish in the English language.

### 2.4. Assessment of Methodological Quality

The study quality of each publication was evaluated, by LF, GMF, AR and VEP, using a 16-item checklist [15]. The quality scores were classified as “low” methodological quality for scores ≤50%, “good” for scores between 51% and 75%, and “excellent” if the score was >75%.

### 2.5. Assessment of Methodological Quality

According to the 16-item checklist Downs and Black modified scale, the average quality score was 91.8%. All studies had excellent methodological quality (quality score > 75%), ranging between 81.25 and 94.7. The inter-rater reliability analysis showed good coherence among the observers, with 0.94 as the kappa value.

## 3. Results and Discussion

The twenty-two selected studies and their main points, analysed in this review, are summarised in Table 1. Subjects with poor motor function showed significant improvements in muscle strength after RMV stimulation [16,17,18,19,20,21,22,23,24,25,26,27,28,29,30,31,32,33] and in power [28,29,34,35,36]. Other less common outcomes expressing an improvement in function are improved joint mobility [37], positive electrophysiological changes in the spinal cord [19], reduced pain caused by insufficient joint stabilisation [23,26,28] and rate of strength development [33]. The after-effects were evident and statistically significant immediately after the end of the treatment. This finding, as well as the long persistence (up to 1 year after treatment), agrees with studies on healthy individuals previously reviewed [4,5,6].

As shown in Table 1, however, in several studies, the improvements have shown their positive effectiveness in tests involving many more muscles than the one treated and in multi-joint tasks. Functional scales [18,20,21,23,24,25,26,27,28,29,30] and digital analysis [16,25,30,32,33,34,35,36] showed complex after-effects. In several cases, the results were able to improve common everyday activities [18,20,24,26,27,28] beyond expectations. Special attention could be paid to the results highlighted in a study [38], not mentioned in the table, showing that stimulatory training applied to the muscles of the floor of the mouth aimed at reducing drooling in children with cerebral palsy. In this case, the significant reduction in drooling was attributed, albeit with indirect evidence, to an improvement in swallowing, a highly coordinated motor task. Finally, one study specifically tested the effectiveness of the proprioceptive stimulation protocol on the training of volleyball players after the seasonal rest break [35]. Their explosive and reactive leg power was assessed at the beginning of the seasonal training and up to 240 days later. Although the athletes followed the same training, the stimulated group showed a much greater improvement than the control group. Interestingly, only 24 h after the end of treatment, the treated athletes showed significant and greater results than their colleagues 240 days later. This finding, discussed later, suggests a prominent role of proprioceptive drive in determining motor efficacy.

In the different studies, improvements persisted until the end of follow-up (up to 12 months after RMV application) without showing a decline. Maintenance was also found in the absence of regular exercise training [17,19,20,22,25,27,29,30,34,36], although the role of associated training seems relevant [34]. Similar long-lasting persistence has been shown in a group of studies [39], not mentioned in Table 1 as they relate to perceptual functions on the recovery of neglect. After a brief and repeated vibratory stimulation applied to the neck muscle related to the perceptual deficit, relief was significant and persistent during the follow-up period (1.5 years) without decay. Another aspect is the diversity of the origin of the deficit in the studies reviewed: there are orthopaedic [16,18,24,33,37], neurological [21,23,26] and post-surgical [16,26,33] conditions, metabolic dysfunctions [32], immobilisation [19] in athletes after prolonged training break or intense physical activities [32,36] and the main cause in Table 1, as well as in society, ageing [17,20,22,27,28,29,30,33,36]. This finding was also emphasised in a recent review [10].

Although the RMV protocols are applied to show a variety of parameters, it is evident (see Table 1) that certain features are dominant. Of the 22 selected studies, 18 report a stimulation frequency between 80–150 Hz. However, 11 [16,18,19,20,24,25,26,30,34,35,36] applied the same protocol (100 Hz, 0.2–0.5 mm, 10 min, 3 applications/day, repeated for 3 consecutive days). On the other hand, only two studies [22,37] applied frequencies lower than this range, and only in two cases [17,27] was the frequency of the FV (Focal Vibration) well above (300 Hz) in the most common range. The positive and lasting after-effects (range 1–12 months), with no signs of decay at the end of the follow-up period, elicited by the shorter but more concentrated protocol, confirm the previous meta-analysis [6].

### 3.1. Bias

Some biases are recognisable among the selected works. The different studies did not use the same stimulation device. Consequently, the relationship between the treated muscle and the device is not well defined and the amplitudes of the vibrations applied are therefore not reliable, even if reported. Furthermore, the role of any specific exercise in the follow-up on the amplitude of the effects and their duration cannot be assessed at present.

### 3.2. Discussion

This review aims to verify the possible role of proprioceptive training in resolving clinical symptoms in negative deficits, as well as in restoring adequate fitness in athletes after an interruption of training (holiday, injury, laziness). The basis of this study is suggested by the results of similar stimulations on healthy individuals [6]. The literature review shows that, as in healthy individuals, conditional abilities, such as strength and muscle power, are improved. Positive, persistent, and consistent results are reported in patients with orthopaedic problems [18,24,36,37], as well as after surgery [16,26] and immobilisation (19). Several studies have described significant improvements in mobility in ageing [17,20,22,27,28,29,30,34,36], in severe neurological diseases [21,23,25,32] and in the recovery of athletes after an injury or a seasonal break [35]. It should be noted that the restoration of physical function was achieved with RFV alone, without other exercise protocols, in 13 out of 22 studies.

The most interesting observations seem to be: (i) despite the diversity of causes of the deficit, trauma, neuromotor impairment, inflammatory conditions, surgical consequences, ageing or sporting activity, the same treatment rapidly improved motor performance; (ii) even if the treatment was the same, improvements were evident in a wide variety of motor tasks; (iii) the after-effects were not limited to the muscle function treated, but involved unexpected body segments and functions; (iv) the positive after-effects did not show decay, even when the observation period lasted months.

#### 3.2.1. Stimulatory Features and Pattern

As pointed out in other reviews [6,10], positive and persistent after-effects were obtained by applying small sinusoidal muscle stretches (<1 mm, preferably between 0.2–0.5 mm) at 100–150 Hz (most commonly at 100 Hz). The low amplitude is recommended to avoid triggering the tonic vibratory reflex that causes muscle fatigue during treatment [40,41]. This muscle lengthening-shortening sequence is typically able to drive the primary spindle afferents [12,13,42] at the same frequency (spindle driving phenomenon).

In healthy individuals, these stimulatory parameters were applied according to two schemes. The first, stereotyped and applied in several studies [16,18,19,20,24,25,26,31,34,35,36], was based on single sessions lasting only 10 min, repeated three times a day, interspersed with 1–2 min of rest. In the other studies, the second scheme applied longer sessions (30–60 consecutive minutes), once a day, variously distributed in time, i.e., one session per day, during 3 or 5 days/week and during 2–26 consecutive weeks. As in healthy individuals, the first protocol, more concentrated and with a frequency of approximately 100 Hz, seems to be more likely to elicit a mechanism like long-term potentiation, capable of supporting a greater persistence of the after-effect.

#### 3.2.2. Relevance of Proprioceptive Activation

Several considerations can be deduced. Among the various studies selected, one aspect is often evident: the short-latency response after the end of the FV treatment (minutes or hours), which suggests a dominant role of the proprioceptive signal in motor efficacy. This finding is evident in the presence of deficits, as well as in the jumping performance of volleyball players [35]. The treated group, after the seasonal rest break, 24 h after the end of FV, developed muscle power corresponding to that achieved by their untreated counterparts after 240 days. This result suggests that the positive after-effects are probably not due to orthopaedic or neurological injury healing. More simply, proprioceptive stimulation improves function. Indirect confirmation of this hypothesis is provided by studies of patients with negative deficits caused by neurological diseases [21,23,25,32]. Two of them were characterised by sensory deficits [25,32], and the others by motor impairments [21,32].

It seems interesting to observe how such interventions act on the proprioceptive system, a crucial target in motor rehabilitation [3]. Persistent and powerful proprioceptive improvements could favour compensatory conditions or trigger virtuous circles, capable of enhancing and ensuring the persistence of positive sequelae. Furthermore, as proposed in the model described above [10], improvements in spatial coordinates could allow for the refinement of developing motor strategies or the rapid development of new motor programmes.

#### 3.2.3. Suggested Mechanism

In previous work, a mechanism of action of RFV was proposed that could lead to lasting improvements in proprioceptive resolution of body position and movement and to more effective motor responses. RFV improves the motor performance of simple and complex functions in healthy individuals and in the presence of motor deficits. The improvement in proprioceptive resolution is consistent with similar after-effects elicited in other sensory systems [11] based on long-term synaptic potentiation. RFV can generate synaptic potentiation in proprioceptive circuits and can shift the activity of neural circuits to a different level to allow the system to be more responsive and adaptive. The potentiation and increased resolution of proprioceptive signals induced by proprioceptive activation [43,44,45,46,47,48,49,50,51,52] may lead to LTP-like plastic changes in cortical and subcortical circuits, promoting more effective motor planning and more accurate spatial coordination [10]. A study on a volleyball team [35] offered indirect evidence for this hypothesis. This observation seems to suggest that proprioceptive training has a much larger and faster effect than exercise-based training, as also highlighted by Filippi et al. [53]. A variety of studies [16,17,19,22,23,26,27,29,32,33,34,35,37] reported similar short-latency results. The possibility of improving both motor execution and spatial perception is also supported, in healthy individuals, by significant after-effects in motor pointing tasks [39,48,54,55].

Finally, both in rehabilitative protocols as well as in sportive physical training, muscle fatigue is a limiting factor. Authors reported, in healthy individuals, an enhancement of fatigue resistance [6] after RFV, which might play a role in the reviewed positive after-effects. The origin of such a relevant result might be in the intense proprioceptive stimulation. It is known that fatigue inhibits proprioceptive signals [56,57] by signals originating from muscle ergoceptive afferents. The effect of fatigue is, therefore, detrimental to adequate motor performance, worsening both muscle function and motor skilling. Literature [58] suggests that proprioceptive activation can exert a reciprocal inhibition on ergoception fatigue signals. Such a mechanism is described as to be like the Gate Control, in which large diameter tactile fibres inhibit smaller diameter pain fibres.

## 4. Conclusions

In rehabilitation, the protocols have to recover the motor capacities impaired, and this is obtained by performing physical training programmes joined with, typically, proprioceptive training and both synergically have the goal of better operating the neural and motor systems together [59]. Even if the intervention requires physical activity, only the proprioceptive system is likewise activated because it is involved with motor planning. For this reason, this aspect would always be taken into account in the rehabilitation protocol to have a more effective and enduring motor recovery.

RFV is demonstrated to be a powerful stimulus of the proprioceptive system and, when administered with proper characteristics at the local level, induces a neural response able to modify the motor drive [6,10]. On this basis, the RFV have the same target as proprioceptive training in rehabilitation. However, as reported above, RFV intervention can enhance motor parameters by itself, without other physical training. Such a possibility could be profitable in case of a low level of the subject’s fitness as well as in the early rehabilitation phases. This is the focal point; RFV is postulated to facilitate body awareness, balance and joint stability as well as the agonist–antagonist management [6,10]. The basal idea is to administer RFV before the traditional intervention, thus enhancing the system responsiveness to the successive elicitation induced by the rehab protocol. The same protocol seems to respond to the need for accelerated rehabilitation and achieving a complete recovery of the coordinative and conditional abilities, not just in athletes [16,17,20,31,33,35,36,59].

## Figures and Tables

**Figure 1 jfmk-08-00103-f001:**
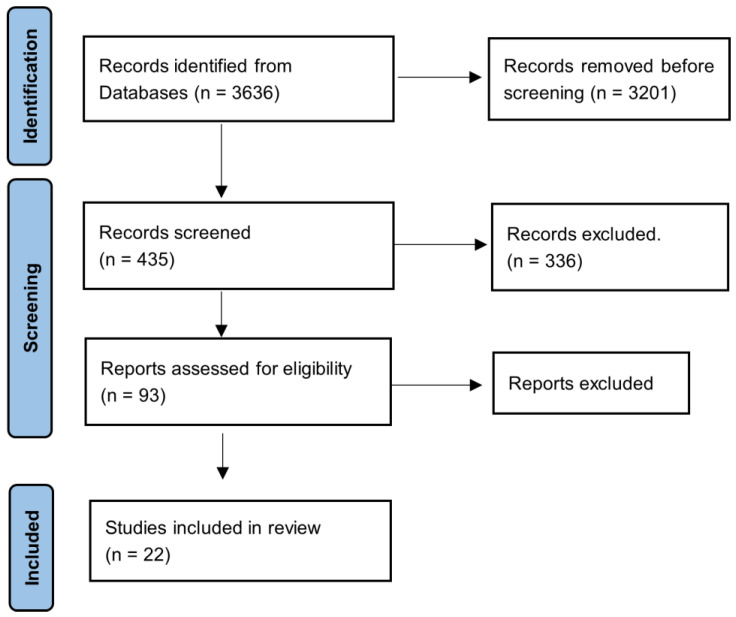
PRISMA flow diagram.

**Table 1 jfmk-08-00103-t001:** Main characteristics of the selected studies included in the review. NR: not reported. The outcome values at the end of the follow-up are reported (* *p* < 0.01; ** *p* < 0.001).

Study	Origin of the Deficit	Sbjts	FV Frequency & Amplitude	Single Application Duration & Repetition	Muscle Body Part Treated/Muscle Contraction	Tests	1st Test and Last Test	Maximal After-Effect
Brunetti et al., 2006 [16]	ACL reconstruction	30	100 Hz; 0.2–0.5 mm	10 min; 3 times a day during 3 consecutive days	Quadriceps/Yes	Stability (cop area, velocity); extensor muscle peak torque	24 h; 270 days	Reduction sway (closed eyes) -40% *; extensor peak force difference vibrated/not vibrated +25% *
Filippi et al., 2009 [34]	Ageing	60	100 Hz; 0.2–0.5 mm	11 min; 3 times a day during 3 consecutive days	Quadriceps/Yes	Stability (cop area, velocity); vertical jump height; muscle power	24 h; 90 days	Power ≈ +50% *; height ≈ +90% *; sway Area ≈ –35% *
Pietrangelo et al. 2009 [17]	Ageing	9	300 Hz; N.R.	15 min; 1–3 times a week for 12 weeks	Quadriceps/No	MVC	Immediately after treatment ending; 16 weeks	MVC ≈ +51% *
Bakhtiary et al., 2011 [37]	Limited hamstring extendibility	30	50 Hz; N.R.	20–60 sec; 3 times a day, 3 times a week for 8 weeks	Hamstring/no	Passive knee extension	Immediately after treatment ending	Knee extension +46% *
Celletti et al., 2011 [18]	Joint hypermobility syndrome	15	100 Hz; 0.2–0.5 mm	10 min; 3 times a day for 3 consecutive days	Quadriceps/Yes	Berg balance scale	10 and 40 days	Berg balance +27% *
Zaho et al., 2011 [19]	Immobilisation	30	100 Hz; 0,3 mm	1 min; 48 times a day for 2 weeks	Soleus/No	V-wave/M-wave	Immediately after treatment ending	Soleus V/M did not change in treated individuals. Untreated showed—30.78% **
Brunetti et al., 2012 [35]	Volleyball players	18	100 Hz; 0.2–0.5 mm	10 min; 3 times a day for 3 consecutive days	Quadriceps/Yes	Explosive and reactive leg power	24 h; 240 days	Treated group explosive leg power +26% **, reactive power +13% **; control group explosive leg power +11% *, reactive power +7.8% *
Tankisheva et al., 2015 [22]	Ageing	50	30–45 Hz; N.R.	30–60 sec; 4–8 times a day for 26 weeks	Quadriceps, Gluteus maximum and medium/No	MVC	Immediately after treatment ending	Quadriceps MVC +13.84% *
Rabini et al., 2015 [24]	Osteoarthritis	50	100 Hz; 0.2–0.5 mm	10 min; 3 times a day for 3 consecutive days	Quadriceps/Yes	WOMAC, SPPB. POMA	3 and 6 months	WOMAC −30% **; SPPB +45% **; POMA +31% **
Celletti et al., 2015 [20]	Ageing	350	100 Hz; 0.2–0.5 mm	10 min; 3 times a day for 3 consecutive days	Quadriceps/Yes	POMA test	1; 6 months	59% of the tested individuals reached the full POMA score **
Brunetti et al., 2015 [36]	Ageing	60	100 Hz; 0.2–0.5 mm	10 min; 3 times a day for 3 consecutive days	Quadriceps/Yes	Stability (cop area, velocity); vertical jump height; muscle power	1; 12 months	Sway −35% **; Vertical Jump + 40% **; Power + 40% **
Ribot-Ciscar et al., 2015 [23]	facio-scapulo-humeral muscular dystrophy	9	80 Hz; 0.5 mm	50 min; A total of 7 sessions, 1 every 4 days	Biceps brachialis; triceps brachialis; pectoralis major/No	Pain analogue visual scale; voluntarily shoulder abduction and flexion maximum amplitudes; MVC	Immediately after treatment ending	Pain analog visual scale, no significant changes; voluntarily shoulder abduction and flexion +20% *; MVC +41% *
Paoloni et al., 2015 [21]	Foot drop	44	120 Hz; 0,001 mm	30 min; 3 times a week, for 12 weeks	Tibialis anterior, peroneus longus/N.R.	Gait analysis	1 month	Improvements in ankle dorsiflexion,
Pazzaglia et al., 2016 [25]	Charcot-Marie-Tooth 1A disease	14	100 Hz; 0.2–0.5 mm	10 min; 3 times a day for 3 consecutive days	Quadriceps/Yes	Berg Balance scale; Dynamic gait index; 6-min walking test; Muscular strength of lower limbs; Body balance; SF-36;	1 week; 1 month	Berg Balance scale +8% *; Dynamic gait index +15% *; =6-min walking test; =Muscular strength of lower limbs; ↑Body balance (Sway path * and velocity *); =SF-36;
Saggini et al., 2017 [27]	Ageing	30	300 Hz; N.R.	15 min; 2 times a week, for 6 months	Trapezius, triceps brachii, latissimus dorsi, rectus abdominis, gluteus maximus, rectus femoris, biceps femoris, and tibialis anterior/N.R.	Hand grip; knee extensores isokinetic strength; POMA test; ECOS-16 questionaire	Immediately after treatment ending	Grip +11% *; Isokinetic strength of the knee extensor +6% *; Poma Test + 5% *; Ecos-16 −17% *
Celletti et al., 2017 [26]	postmastectomy recovery	14	100 Hz; 0.2–0.5 mm	10 min; 3 times a day for 3 consecutive days	Pectoralis minor and the biceps brachi/Yes	DASH; questionnaire, Body Image Scale, McGill pain questionnaire, Constant Scale, and Short Form 36 questionnaire.	Immediately after treatment ending	DASH scale −28% *; Constant scale +14% *; theMcGill pain questionnaire −23% *; ↑Short Form 36 questionnaire (=physical mental score)
Benedetti et al., 2017 [28]	Ageing	30	150 Hz; N.R.	20 min; Once a day through five consecutive days, for 2 consecutive weeks	Rectus femoris, vastus medialis, and vastus lateralis	WOMAC; VAS; STAIR CLIMBING; TUG	48 h	WOMAC −20% **; VAS −49% **; STAIR CLIMBING −13% **; TUG −11% **
Souron et al., 2018 [29]	Ageing	17	100 Hz; 1 mm	1 h; 3 times a week, for 4 weeks	Rectus femoris/No	MVC, Vertical jump performance	Immediately after treatment ending	MVC ≈ +11% *; Maximal jump heights SJ ≈ +15.2% *, CMJ ≈ +6.5% *
Iodice et al., 2019 [31]	Athletes’ effects of eccentric exercise	30	120 Hz; 1,2 mm	15 min; once	Vastus intermedius, rectus femoris, vastus lateralis, vastus medialis, gluteus maximus, biceps femoris, adductor longus and magnus	isokinetic evaluation, stabilometric test, perceived soreness evaluation	48 h	MVC ≈ +13% **
Attanasio et al., 2020 [30]	Ageing	30	100 Hz; 0.2–0.5 mm	10 min; 3 times a day for 3 consecutive days	Quadriceps/Yes	Body balance, POMA test, TUG test	1 week	Sway ≈ −27% *; POMA test ≈ +20% **; TUG: rotation speed ≈ +8% **; duration ≈−19% *, standing up ≈ −13% **
Rippetoe et al., 2020 [32]	Diabetic Peripheral Neuropathy	23	120 Hz; 1.2 mm	10 min; 3 times a week, for 4 weeks	Tibialis anterior, quadriceps, and gastrocnemius/No	Gait Analysis	Immediately after treatment ending	↑Gait speed *, ↑cadence *, ↑stride time *, ↑left and right stance time *, ↑duration of double limb support *, ↑left and right knee flexor moments*
Coulandre et al., 2021 [33]	ACL reconstruction	30	100 Hz; 1 mm	1 h; only once	Quadriceps/No	MVC Rof force development	Immediately after treatment ending	Force decrease in vibrated subject −50% then unvibrated participants

## Data Availability

PubMed, Web of Science and The Cochrane Library.

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
