# Peer review of "Effectiveness of Focal Muscle Vibration in the Recovery of Neuromotor Hypofunction: A Systematic Review"

_jfmk, 2023, doi:10.3390/jfmk8030103_

Round 1
Reviewer 1 Report
This is a very interesting study aimed at investigating the effect of the proprioceptive stimulation exerted by focal muscle vibration on poor motor function.
The study results show that RMV involves a powerful stimulus of the proprioceptive system able to modify the motor drive and improve motor deficit through positive and long lasting after effects. that were not limited to the muscle function treated, and did not show decay, even when the observation period lasted months.
Strengths:
Even though on the one hand including different causes of motor deficit (i.e. trauma, inflammatory conditions, surgical consequences and even ageing or sporting activity) could make drawing robust conclusion more difficult, on the other hand it shows that the same treatment can rapidly improved motor performance regardless the underlying pathology. In my opionion this contributes to make the suggested RMV mechanism (i.e. proprioceptive processing conditioning) more plausible.
It is also worth noting that this study also review the motor after effects that goes beyond the simple strength improvement, showing that RMV could promote more effective motor planning and more accurte spatial coordination.
Finally, since it is widely reported that RMV’s capability of inducing synaptic plasticity using the proprioceptive pathway depends on vibration parameters, I also appreciated discussion of this aspect reviewing difference between the analyzed treatment protocols regarding the stimulation frequency, the vibration amplitude and the avoidance of TVR elicitation.
Thus, I have just some minor points. In my opinion, the paper would be considerably improved by reviewing the following minor issues:
- I my opinion, the major flaw is that talking about “recovery of poor motor function” (title) or “stable negative deficits of muscle function” (inclusion criteria) make the review topic ambiguous and deserves to be clarified both in the title and in the inclusion criteria. For exemple, since stroke too often leeds to “stable motor deficit”, I expected from the title that the review would also cover stroke, but only one study on chronic stroke patients was included. And the same could be said regarding “poor motor function” with stable negative deficit due to others neurological diseases such as PD (akinetic-rigid sub-type), spasticity in multiple sclerosys or stroke and so on. So, please, review the title and clarify selection criteria.
- Synaptic potentiation in proprioceptive circuits does not represent the sole RMV action mechanism. Specifically, hyperactivation of muscle proprioceptors induced by RMV can also produce LTP-like plastic changes in the sensorimotor cortex and modulate its connections to the spinal cord. Thus, the LTP-like plastic changes induced by fMV do not involve the sole proprioceptive circuits but probably entail a whole motor network relearning achieved through the plasticity-based modulation of the effective connectivity. In this regard, please breafly discuss also evidence on LTP-like plastic changes induced in the brain by fMV through operant conditioning of cortical excitability and perfusion as reported by a great amount of neurophysiological and neuroimaging studies.
Author Response
- I my opinion, the major flaw is that talking about “recovery of poor motor function” (title) or “stable negative deficits of muscle function” (inclusion criteria) make the review topic ambiguous and deserves to be clarified both in the title and in the inclusion criteria. For exemple, since stroke too often leeds to “stable motor deficit”, I expected from the title that the review would also cover stroke, but only one study on chronic stroke patients was included. And the same could be said regarding “poor motor function” with stable negative deficit due to others neurological diseases such as PD (akinetic-rigid sub-type), spasticity in multiple sclerosis or stroke and so on. So, please, review the title and clarify selection criteria.
The review is limited to studies of individuals with neuromuscular hypofunction, without spasticity, dyssynergy, and dystonia. This limitation is justified by the need to test the effect of focal vibration in individuals in whom motor activity should be enhanced, consistent with the effect observed in healthy subjects. Therefore, subjects with hypertonia due to different levels of circuit excitability were excluded. For this reason, studies on stroke causing spasticity were not included in the analysis. However, it is difficult to change the title accordingly to show that the review was limited only to individuals with impaired motility without spasticity.
The title has been changed by replacing "poor motor function" with "neuromotor hypofunction." But we think it is still ambiguous. We do not have different solutions.
We added two phrases in “Introduction” and in “Methods”
Introduction
“The appraisal is limited to studies of individuals with neuromuscular hypofunction without spasticity, dyssynergy, and dystonia. This limitation is justified by the desire to test whether the effect of focal vibration in individuals, in whom motor activity should be enhanced, is in line with the effect observed in healthy subjects. Therefore, subjects with hypertonia were excluded, due to different levels of their circuit excitability.”
Methods
Exclusion criteria were absence of spasticity, rigidity, dystonia, dyssynergia.
We hope that the reason for our choice is now clear in the manuscript.
- Synaptic potentiation in proprioceptive circuits does not represent the sole RMV action mechanism. Specifically, hyperactivation of muscle proprioceptors induced by RMV can also produce LTP-like plastic changes in the sensorimotor cortex and modulate its connections to the spinal cord. Thus, the LTP-like plastic changes induced by fMV do not involve the sole proprioceptive circuits but probably entail a whole motor network relearning achieved through the plasticity-based modulation of the effective connectivity. In this regard, please breafly discuss also evidence on LTP-like plastic changes induced in the brain by fMV through operant conditioning of cortical excitability and perfusion as reported by a great amount of neurophysiological and neuroimaging studies
Regarding the second point, we agree with the referee that plastic events in the cortex were not emphasized in this paper. This was suggested in our previous work (frontiers). We have now included a sentence about the possible effect of proprioceptive enhancement on the cortical plasticity and the evidence of plastic events in a large area of the brain.
The new sentence:
“The potentiation and increased resolution of proprioceptive signals induced by proprioceptive activation [44-53] may lead to LTP-like plastic changes in cortical and subcortical circuits, promoting more effective motor planning and more accurate spatial coordination [10]”.
Reviewer 2 Report
This is very interesting study with potential clinicla application. I think, that some statistical comaprison of results from previous studies may significantly improve this review. I advice that , the authors should add these results from the statistical analyys. The cocncusions should be strengthened by the new results . Potential clinical aplications should be also added and addressed.
Author Response
This is very interesting study with potential clinicla application. I think, that some statistical comaprison of results from previous studies may significantly improve this review. I advice that , the authors should add these results from the statistical analyys.
We completely agree with Referee 2’s observation, in fact, our initial idea was to perform a meta-analysis of the literature. However, we were forced to give up because the samples, initial and experimental conditions were too different among the different available studies.
The cocncusions should be strengthened by the new results . Potential clinical aplications should be also added and addressed.
Concerning the interesting of this findings are already underline in the final part of the manuscript.